# Highly scalable photoinduced synthesis of silanols via untraversed pathway for chlorine radical (Cl•) generation

Argha Saha[1,3], Wajid Ali[1,3], Daniel B. Werz[2] ✉ & Debabrata Maiti [1] ✉

The emergence of visible light-mediated synthetic transformations has transpired as a promising approach to redefine traditional organic synthesis in a sustainable way. In this genre, transition metal-mediated photoredox catalysis has led the way and recreated a plethora of organic transformations. However, the use of photochemical energy solely to initiate the reaction is underexplored. With the direct utilization of photochemical energy herein, we have established a general and practical protocol for the synthesis of diversely functionalized organosilanols, silanediols, and polymeric siloxanol engaging a wide spectrum of hydrosilanes under ambient reaction conditions. Streamlined synthesis of bio-active silanols via late-stage functionalization underscores the importance of this sustainable protocol. Interestingly, this work also reveals photoinduced non-classical chlorine radical (Cl•) generation from a readily available chlorinated solvent under aerobic conditions. The intriguing factors of the proposed mechanism involving chlorine and silyl radicals as intermediates were supported by a series of mechanistic investigations.

Silanols constitute a myriad of building blocks for polymeric materials[1]. Also, silanols are equally known as cross-coupling partners in metal-catalyzed reactions[2], directing auxiliary for C−H functionalization[3], organocatalysts for asymmetric reactions[4] and isosteres of bio-relevant molecules[5] (Fig. 1a). Apart from being precursors in different organic transformations, application of silanols in material and medicinal science inspired scientific community to design protocols with the focus on practicality, scalability, and sustainability[6]. For low molecular weight silanols, hydrolysis of chlorosilanes is a straightforward approach[7]. However, the requirement of precise reaction conditions and non-applicability for sterically hindered silanols also prone to condensation to form disiloxanes limits its applications. Direct oxidation of Si–H bonds of hydrosilanes also employed for the synthesis of a wide range of silanols. Nevertheless, oxidation of Si−H bonds requires a stoichiometric amount of strong oxidizing agents such as peracids, permanganate, ozone, osmium tetraoxide as well as toxic metal salts (Fig. 1b)[8–12]. Also, selectivity, limited substrate scope, low functional group tolerance, and oxidation reactions often

lead to undesired byproducts like disiloxanes thus limiting their broad applications. Several transition metal catalysts such as Ti, Re, W, Co, and Mn in conjunction with oxidant ($H_2O_2$ or $O_2$) were also exploited to convert hydrosilanes to silanols[13–15]. Utilization of water as an eco-friendly oxygen source a series of catalytic systems using noble metal catalysts such as Pd, Cu, Ru, Ir, Ag, Pt, Au, and Rh were developed to convert hydrosilanes to silanols (Fig. 1b)[16–21]. Despite these most of the metal catalysts used are expensive and commercially unavailable and also resulted in the formation of disiloxanes as the byproduct. To further cut down the limitation of transition metal catalysis organocatalytic[22], enzymatic[23–25], and recently electrochemical approaches[26] have been explored for selective oxidation of hydrosilanes to silanols in the presence of $H_2O_2$, $O_2$, or $H_2O$ as the oxygen source. Although, these methods provide a greener approach but has its own limitations. Hence, it would be fascinating and yet challenging to develop a mild and practical protocol that allows highly selective oxidation of hydrosilanes to silanols under ambient conditions.

[1]Department of Chemistry, Indian Institute of Technology Bombay, Mumbai, India. [2]Albert-Ludwigs-Universität Freiburg, Institute of Organic Chemistry, Albertstr. 21, 79104 Freiburg, Germany. [3]These authors contributed equally: Argha Saha, Wajid Ali. ✉e-mail: daniel.werz@chemie.uni-freiburg.de; dmaiti@iitb.ac.in

Chlorine radical (Cl·) has been utilized immensely for the generation of a carbon-centred radical[27] of the C($sp^3$)−H bond via hydrogen atom transfer (HAT)[28]. There is a growing demand to develop an innovative alkylation reaction involving these chlorine radical initiators. Recently the photo-electrochemical cross-coupling reaction showed anodic oxidation of tetraethyl-ammonium chloride (TEACl) to deliver chlorine radical (Cl·) (Fig. 1c)[29]. Similarly, tetrabutyl-ammonium chloride (TBACl) can serve as a chlorine radical source under photo-irradiation in the presence of CeCl$_3$ for a cross-coupling reaction (Fig. 1c)[30]. The fundamental aspect of these processes evolves around the catalytic generation of chlorine radical (Cl·) via the oxidation of chloride salts. Another class of

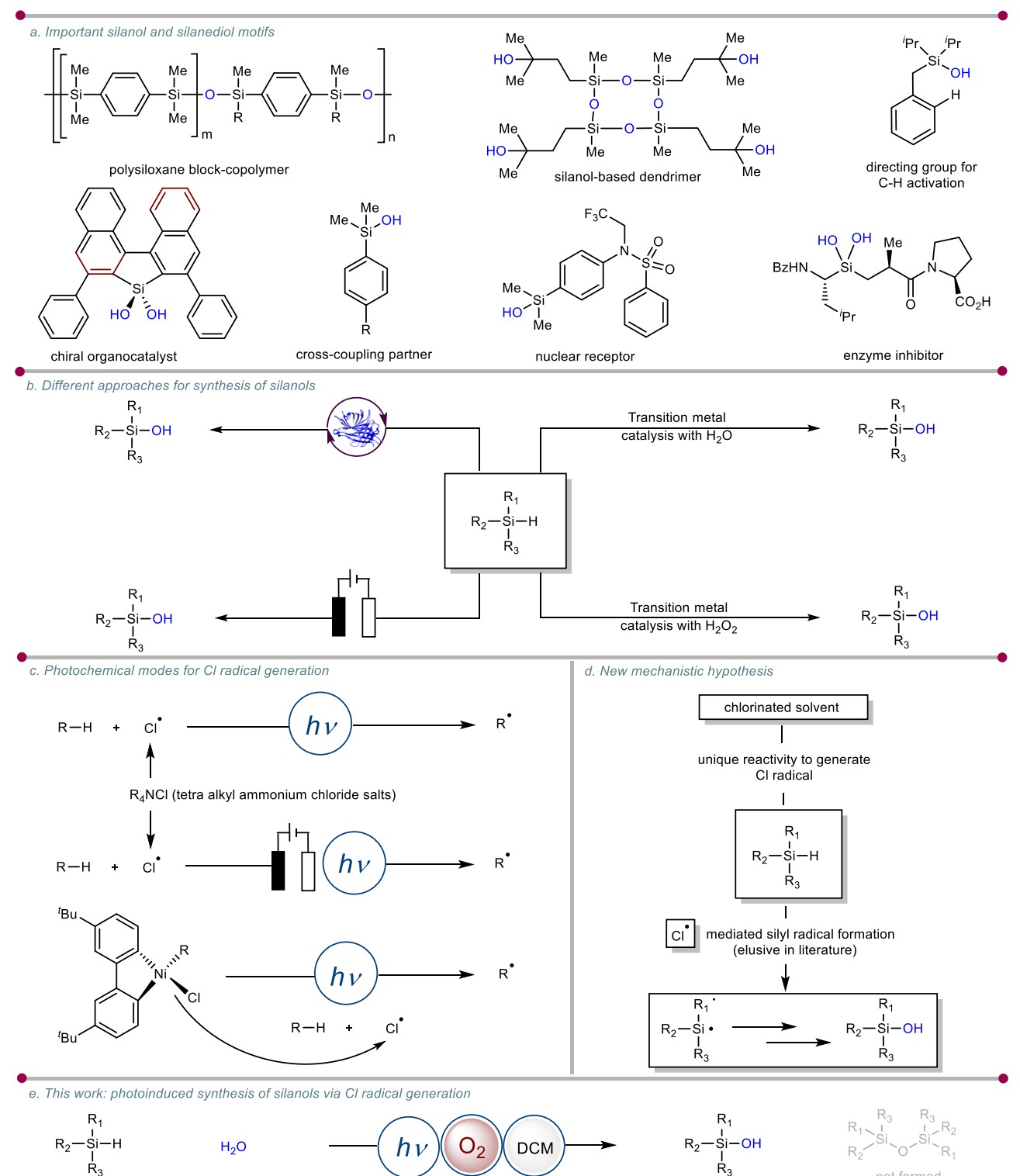

**Fig. 1 | Photoinduced synthesis of silanols via non-traditional Cl radical generation. a** Representative example of the important silanol and silanediol motifs. **b** Different approaches for the synthesis of silanols. **c** Photochemical modes for Cl radical generation. **d** New mechanistic hypothesis. **e** Photoinduced synthesis of silanols via Cl radical generation (this work).

**Table 1 | Optimization of photoinduced silanol synthesis**

| Entry | Deviation from standard conditions | Yield (%)[a] |
|---|---|---|
| 1 | None | 95 |
| 2 | In dark | NR |
| 3 | White LEDs instead of blue LEDs | 14 |
| 4 | Air instead of $O_2$ | 56 |
| 5 | $N_2$ instead of $O_2$ | < 15 |
| 6 | Dichloroethane instead of dichloromethane | 85 |
| 7 | Chlorobenzene instead of dichloromethane | 26 |
| 8 | Bromobenzene instead of dichloromethane | 5 |
| 9 | $CH_3CN$ instead of dichlorobenzene | 11 |
| 10 | 2.0 equiv. of $H_2O$ instead of 5.0 equiv. | 78 |
| 11 | 10.0 equiv. of $H_2O$ instead of 5.0 equiv. | 93 |
| 12 | 12 h instead 28 h | 73 |

[a]Yield determined by $^1$H NMR using TMB (trimethoxy benzene) as internal standard.

chemistry for the generation of chlorine radical (Cl•) includes photolysis of Ni(III) aryl chloride species, which is generated by single-electron oxidation of a typical Ni(II) intermediate (Fig. 1c)[31]. Instead of readily available inorganic salts, this method restrained its applicability only to the alkyl radical. The implementation of chlorine radical (Cl•) to generate heteroatom-centered radicals remains a major challenge in organic chemistry. Herein, we developed a protocol that demonstrated the unique example of a chloride salt-free method to generate chlorine radical (Cl•) from dichloromethane under visible light conditions (Fig. 1e). We envisaged that chlorinated solvent under photochemical conditions would provide the rudimentary chlorine radical (Cl•)[30,32]. Then, the chlorine radical (Cl•) would furnish heteroatom-substituted free radical from hydrosilanes which ultimately deliver the desired silanol product with the help of water (Fig. 1d). Consequently, our method has immense potential to exhibit a broad substrate scope of silanols with high functional-group compatibility and it is applicable to late-stage functionalization of complex molecules as well as preparation of drug building blocks.

## Results

### Reaction optimization

Based on the above mentioned mechanistic hypothesis, we initiated our investigations with tris(trimethylsilyl)silane as a model substrate in DCM (dichloromethane) under visible light conditions. The outcome of the preliminary reaction was encouraging as the desired tris(trimethylsilyl)silanol was obtained in a moderate yield of 23%; indicating the result is in line with the mechanistic postulate. Subsequently, the reaction was performed under an oxygen atmosphere and the outcome of the reaction shows a significant increase in yield (see Supplementary Information for optimization details, Section 2.2). With this condition, we noticed that the inclusion of water helped accelerate the transformation. After that, we varied the amount of water and 5 equiv. turned out to be optimum. To understand the role of chlorinated solvents, we explored different chlorinated solvents including dichloroethane, chlorobenzene, and carbon tetrachloride. Among these, dichloromethane emerged as the most suitable medium. The aforementioned condition was not amenable to other halogenated solvents like bromobenzene and fluorobenzene. Notably, the yield of the desired product was increased when the reaction was kept for 28 h. However, a further increase in either water amount or reaction time does not improve the reaction efficiency (see Supplementary Information for optimization details, Section 2.2). A brief optimization

of the reaction conditions is given in Table 1. Finally, after a series of optimizations, we accomplished a reaction condition that affords 93% of the isolated desired (TMS)₃SiOH product.

### Scope of the methodology

This discovery of a benign protocol established an ideal stage for interrogating the process in terms of scope and applicability. The dialkyl(aryl)silanes with different electronic and steric substitution patterns were explored initially (Fig. 2). Functional groups including alkyl (1–3, 5–6), fluoro (4), thiomethyl (7), trifluoromethyl (8), and chloro (9) (Fig. 2) were found compatible. Irrespective of the 1,2- and 3,5-di-substitution pattern of the arenes, the photoinduced transformation occurred smoothly with excellent yields of their corresponding products (10–11) (Fig. 2). Apart from phenyl rings, silanol with polyaromatic anthracene core was also synthesized in good yields (12) (Fig. 2). Further, sterically hindered diphenyl (13), triphenyl silanol (15), as well as bulky diphenyl alkoxy silanol (14), were prepared in excellent yields (Fig. 2). However, due to the steric bulk of these silanes it is difficult to hydrolyze them and harsh conditions are needed. But with our present protocol, we can easily hydrolyze them in a sustainable way. Following this, we choose a series of benzylic silanes to showcase the remarkable compatibility of this protocol. These benzyl silanes exhibited excellent functional group tolerance with various substituents (16–27), further demonstrating the high chemo-selectivity of our method (Fig. 2). Notably, benzylic C–H bonds are susceptible to oxidation under previously reported oxidative conditions[33]. The long-chain bulky silane can also be converted into its silanol analog (28) with this method. It is worth noting that alkenyl- (29) and alkynyl-substituted (30) hydrosilanes were also transformed into silanols (Fig. 2). Remarkably, epoxidation of a C = C moiety was not observed under our present reaction conditions[34]. The heteroaromatic hydrosilanes such as thiophene, benzothiophene, benzofuran and indole demonstrated as exciting substrates in this photoinduced strategy, leading to the targeted products (31–34) in synthetically useful yields (Fig. 3). Subsequently, long-chain aliphatic silanes which are considered to be difficult substrates were also transformed to functionalized silanols (35–40) (Fig. 3). Siloxanols are very important compounds in terms of their practical applications. However, methods to synthesize siloxanols have remained elusive. Under our developed reaction conditions, functionalized siloxanol (41) can be synthesized in an efficient manner (Fig. 3). Geminal silanediols are another class of structural building blocks present in many drugs, organocatalysts, and silicon-based materials[1–5]. The catalytic oxidation of dihydrosilanes to silanediols containing two Si–H bonds is always a challenging task. Hence, we showcased the highly selective synthesis of alkyl (42) and aromatic (43) geminal silanediols using this methodology (Fig. 3). A representative example of dihydrodiphenylsilane was also shown, which can be successfully converted to silanediol (44) in 60% yield. To demonstrate the amenability of this photochemical protocol, late-stage modification of a series of natural products and drug derivatives was examined. Sartanbiphenyl (OTBN), which is one of the key pharmaceutical precursors for the formal synthesis of drugs such as losartan, valsartan, Irbesartan, and olmesartan, could be transformed into the corresponding silanol analog in synthetically useful yield (45) (Fig. 3). The newly developed photochemical system expedites the synthesis of menthol (46) as well as loxoprofen (47) and ibuprofen (48) based silanols in moderate yields (Fig. 3), creating an expanded chemical space. From the pharmacokinetics and pharmacodynamics of biological studies, it has been evaluated that the drug candidate is ameliorated by the replacement of a carbon center with silicon or by harnessing the "Silicon Switch"[35]. Eventually, these pharmacologically important hydrosilanes are also converted into their corresponding silanol analogs

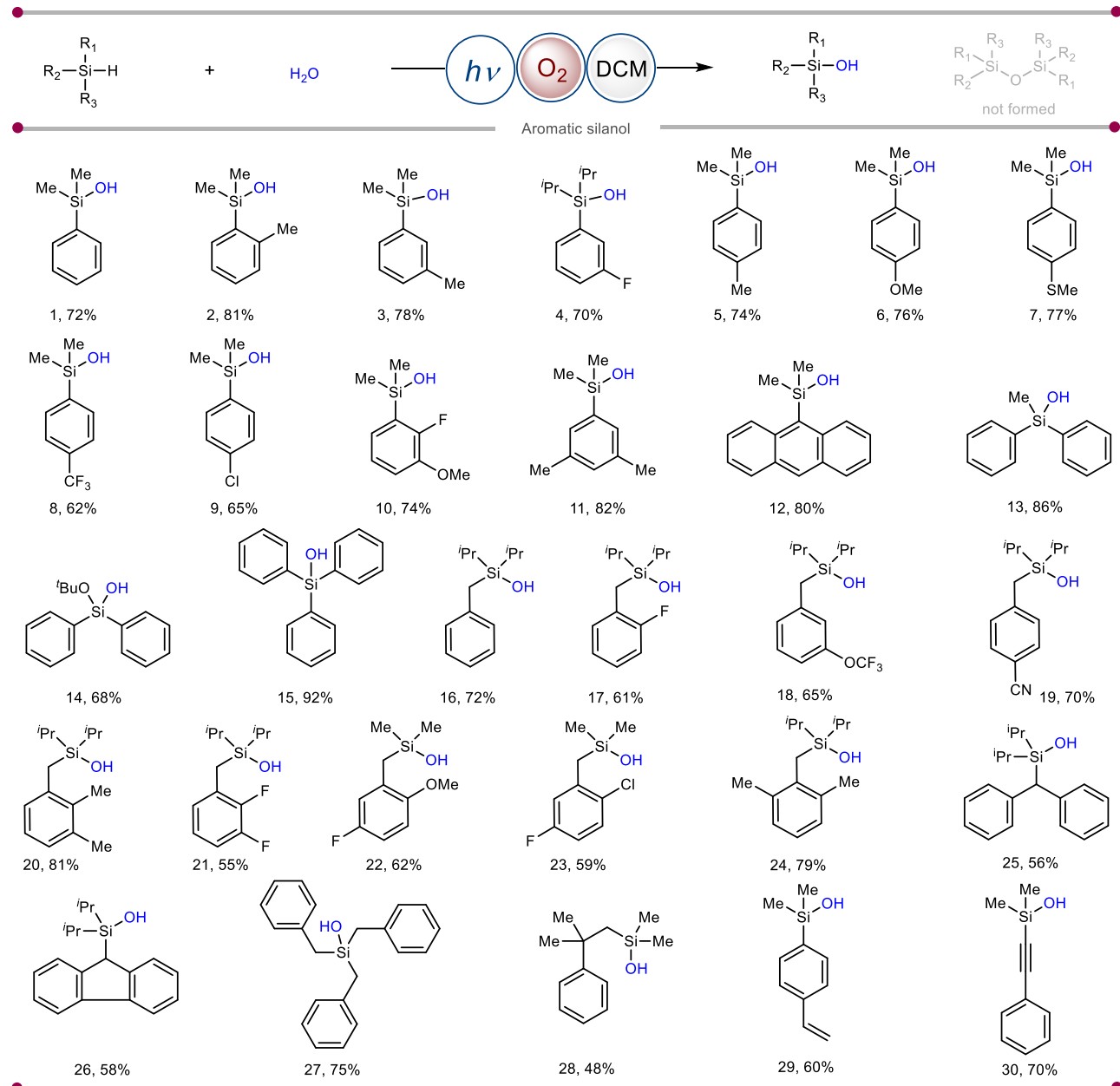

**Fig. 2 | Scope of aromatic silanol synthesis.** Reaction conditions: silane (0.2 mmol), H$_2$O (5.0 *equiv.*), O$_2$, DCM (1 mL), blue LEDs, 28 h. Isolated yields are reported. For experimental details, see Supplementary Methods.

(**49–50**) (Fig. 3). These silanol products with elaborate molecular architectures could have potential applications in medicinal chemistry. Remarkably, for all the substrates this photoinduced reaction proceeds without the formation of disiloxane as a byproduct.

After realizing the generality and compatibility aspects of the photo-mediated protocol, we were excited to demonstrate the scalability of the transformation. With this aim, we employed continuous-flow chemistry. The tubular microreactor merged with blue LED has a high surface-to-volume ratio. These features ideally facilitate the influx of photons in photochemical transformations[36]. The microreactor was connected via PTFE tubing and a blue LED was placed above the microreactor to cover the maximum surface area. The reactants were provided by a syringe pump. With this setup, we first studied the continuous flow reaction on tris(trimethylsilyl) silane which was also used during the optimization for batch reaction. Notably, the reaction time for the batch method was 28 h,

while the same under-flow set-up gave quantitative conversion in 1 h residence time (see Supplementary Information for optimization details, Section 2.2). Overall, the continuous-flow reaction setup allows the photoinduced synthesis of organosilanols at a faster rate with an increase in productivity by many folds. Under this modified reaction condition, silanol **35** is produced at 44 g/h/L and up to 1.05 kg/day/L reactor volume. Thereafter, we proceeded to intensify the productivity of other substituted aliphatic and aromatic silanols. This flow setup design proved highly efficient and allowed a diverse range of silanol products (**51–59**) (Fig. 4). The products **52**, **53**, and **56** were also scaled up to 32.4, 37.4, and 41.4 g/h/L, respectively. Following this methodology, silanols can be produced with high efficiency in a continuous-flow fashion, which now overcomes the limitations of photochemical reactions in conventional batch systems. Thus, continuous flow merged with the photochemical oxidation of silanes renders scalability and sustainability of the overall transformation[37].

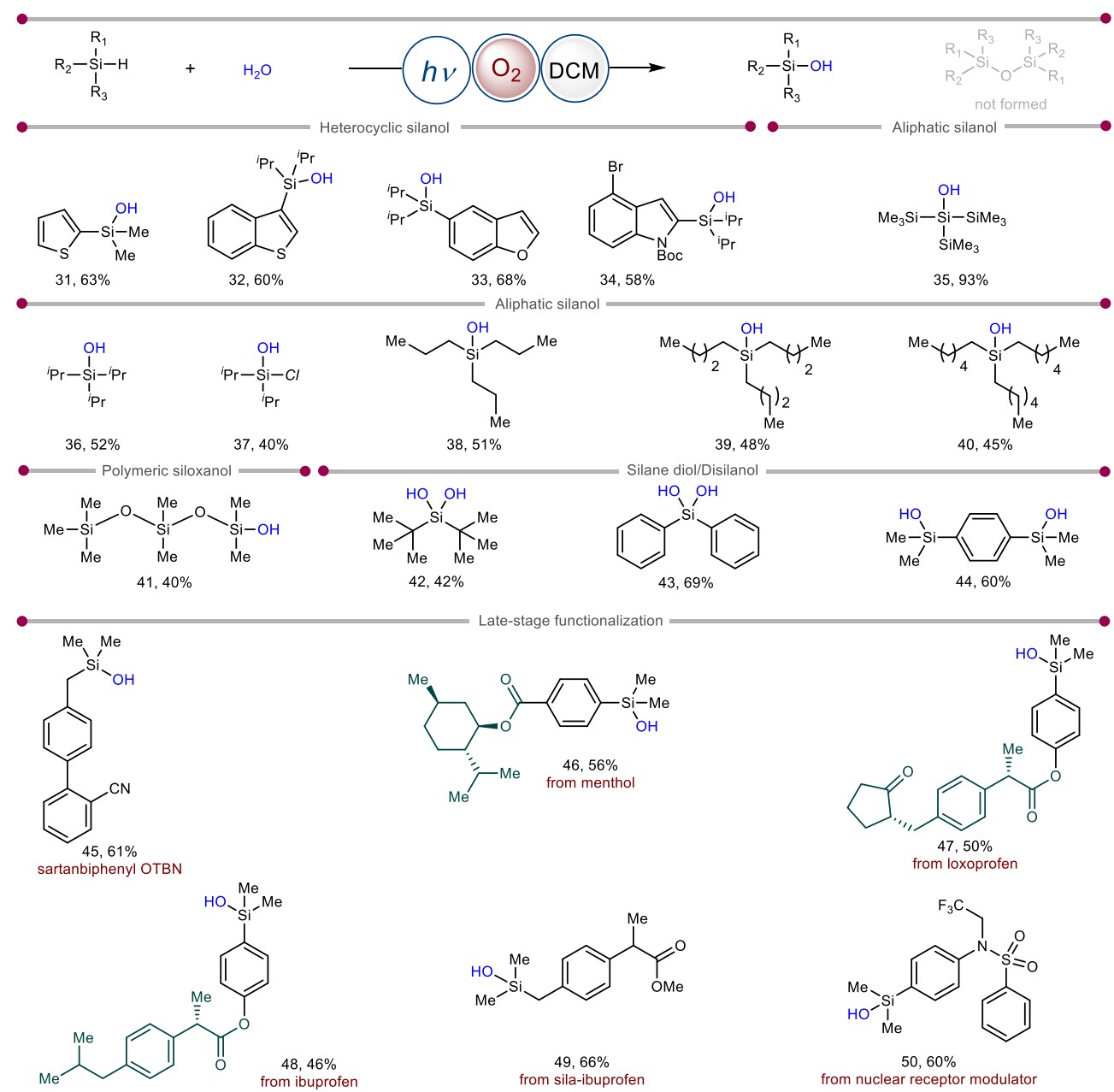

**Fig. 3 | Scope of heterocyclic, aliphatic, polymeric, and late-stage diversification of silanes.** Reaction conditions: silane (0.2 mmoL), H₂O (5.0 *equiv*.), O₂, DCM (1 mL), blue LEDs, 28 h. Isolated yields are reported. For experimental details, see Supplementary Methods.

## Application

The transformable nature of the silanol moieties led us to conduct a series of further manipulations, thus allowing facile access to value-added functional group transformation (Fig. 5). In a semi-one pot manner, product **3** afforded compound **60** in 42% overall yield (Fig. 5)[38]. The palladium-catalyzed cross-coupling of (4-methoxyphenyl)-dimethylsilanol with methyl 4-iodobenzoate gave the corresponding biaryl product **61** in excellent yield (Fig. 5)[39]. The photoinduced olefination of product **16** acting as a weak coordinating directing group delivered the *ortho* olefinated compound 62 (Fig. 5)[40] in good yield. Product **56** provided oxasilacycle **63** (Fig. 5)[38] via the Pd-catalyzed directed aromatic C−H oxygenation reaction (see Supplementary Information for more details, Section 2.4). Products **42** and **43** upon treatment with germanium chlorides transformed into protective coating materials were readily used in industries[i]. reaction predominantly with the nitrile functionality to afford the oxazole **54** with a

trace amount of cyclopropane product formation; which again highlights the chemoselectivity aspect of the transformation.

## Mechanistic studies

To understand the role of visible light, the reaction was performed in the absence of it, giving no product in the present transformation. (Fig. 6a, Eq. 2). This experiment clearly suggests the imperative role of light in the initiation of the reaction. When the same reaction was performed under an argon atmosphere, a trace amount of product was observed (Fig. 6a, Eq. 1, see Supplementary Information for more details, Section 3.1). To validate the superoxide formation, a reaction was conducted with potassium superoxide in an argon atmosphere. A yield of 62% confirmed the in situ generation of superoxide in the reaction medium (Fig. 6b, Eq. 1)[41]. The superoxide formation also relied on irradiation of visible light. So, a control experiment was performed with potassium superoxide in an argon atmosphere without blue LEDs

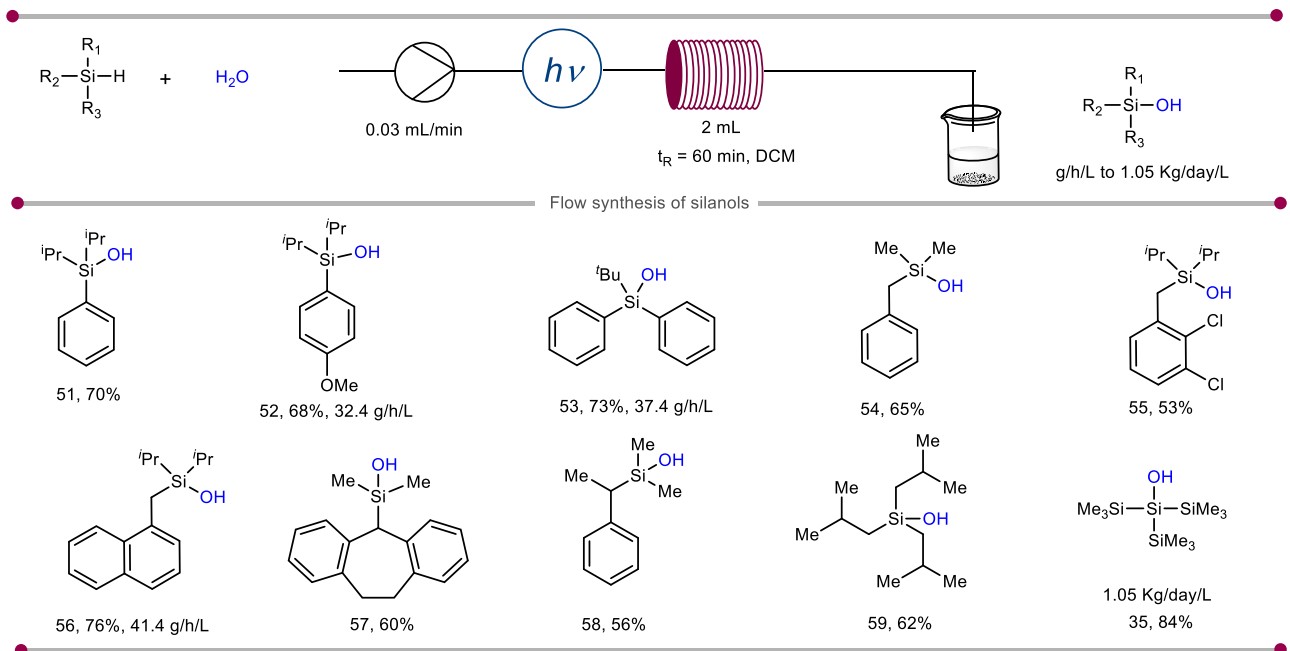

**Fig. 4 | Utilization of photochemical flow chemistry for highly scalable silanol synthesis.** Reaction conditions: silane (0.2 mmoL), $H_2O$ (5.0 *equiv.*), $O_2$, DCM (1 mL), blue LED, 1 h. Isolated yields are reported. For experimental details, see Supplementary Methods.

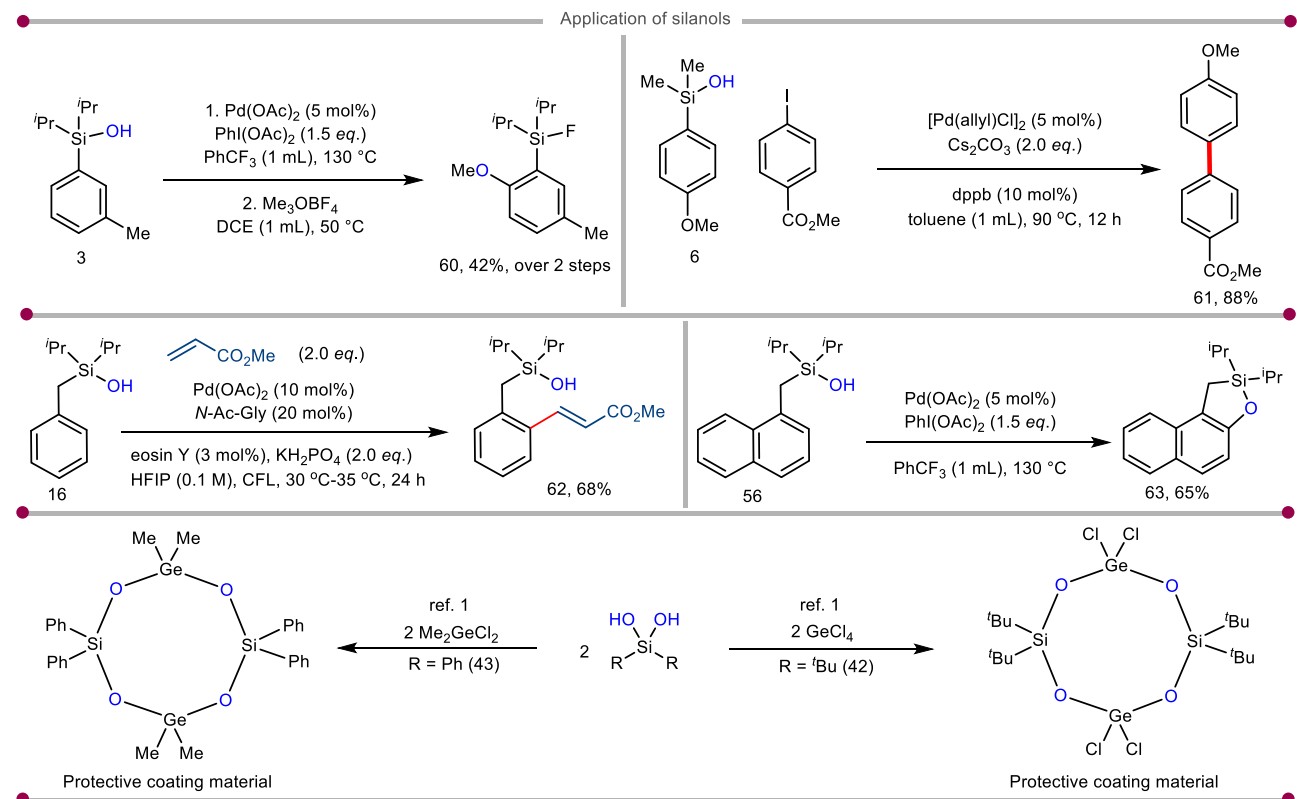

**Fig. 5 | Applicative potential of the protocol.** Isolated yields are reported. For experimental details, see Supplementary Methods.

for 28 h. It is observed that there is no product formation. Hence, it can be concluded that light is necessary for the superoxide formation (Fig. 6b, Eq. 2, see Supplementary Information for more details, Section 3.2). Another key feature of this protocol is the initiation of the first step of the reaction. It is well documented in the literature that visible light has the capability to generate singlet oxygen species from molecular oxygen[42]. In order to confirm the presence of singlet oxygen, we conducted a trapping experiment with anthracene[43] as the trapping agent. This resulted in the formation of the endoperoxide product (Fig. 6c, see Supplementary Information for more details, Section 3.3). Importantly, this endoperoxide product was not observed in reactions conducted in dark conditions. This provides compelling

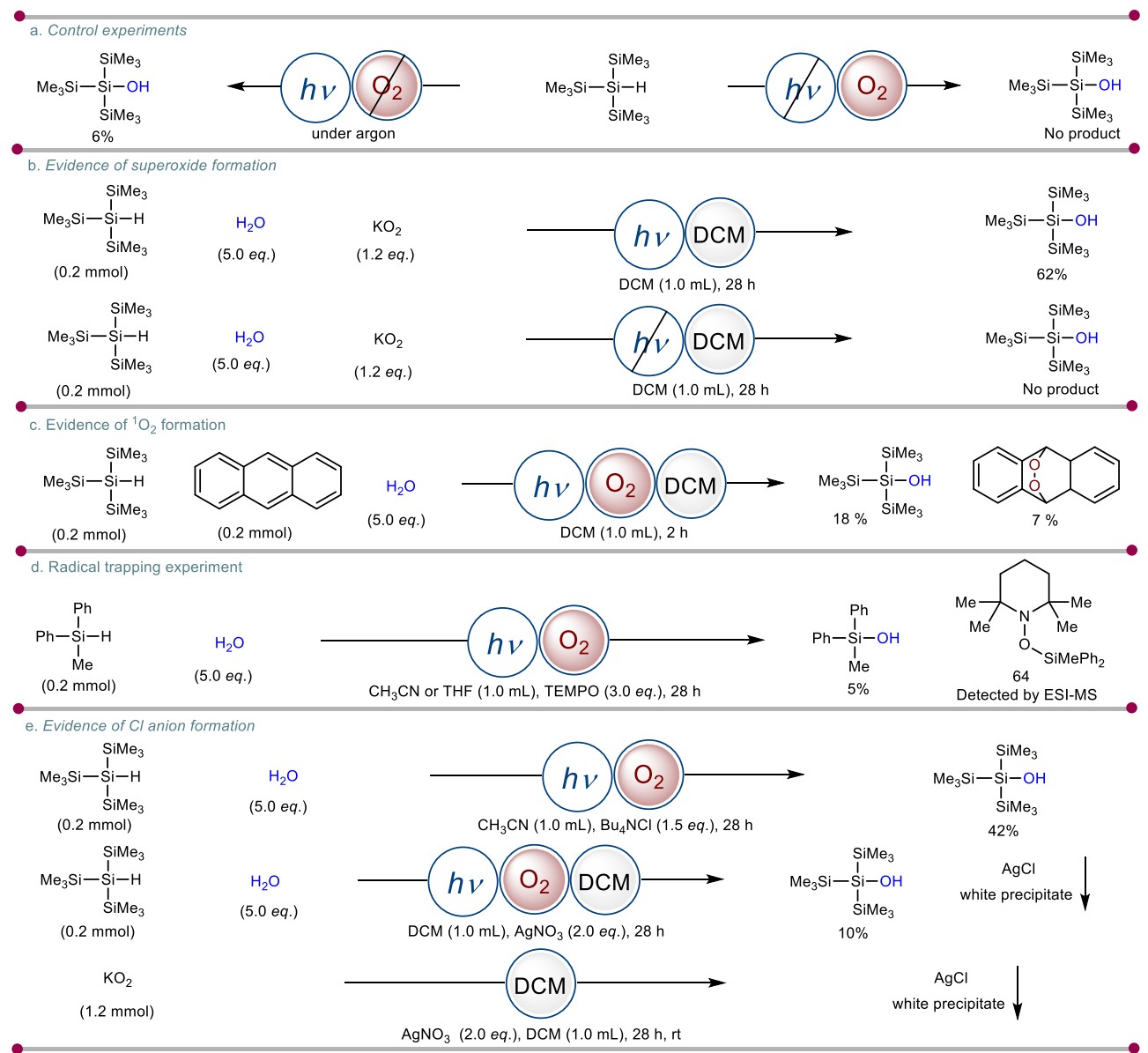

**Fig. 6 | Mechanistic aspects of photoinduced silanol synthesis.** [a]Reaction conditions: **a** control experiments. Eq. (1): silane (1.0 mmoL), $H_2O$ (5.0 *equiv.*), $O_2$, DCM, 28 h. Eq. (2): silane (1.0 mmoL), $H_2O$ (5.0 *equiv.*), argon, DCM, blue LEDs, 28 h. **b** Evidence of superoxide formation. **c** Evidence of $^1O_2$ formation. **d** Radical trapping experiment. **e** Evidence of Cl anion formation. For experimental details, see Supplementary Methods.

evidence supporting the production of singlet oxygen ($^1O_2$) within our reaction system. After that, the formation of silyl radical was further trapped with 2,2,6,6-tetramethylpiperidin-1-oxyl (TEMPO) adduct referring that the radical reaction to be operative (Fig. 6d, see Supplementary Information for more details, Section 3.4). To understand the fundamental role of chlorinated solvent in initiating chlorine radical (Cl·) formation, a series of control experiments were performed. At first, dichloromethane was displaced with common inorganic chloride salts to induce chloride anion by taking acetonitrile as the reaction solvent. The reaction proceeds with 42% yield indicating the formation of Cl anion in the catalytic cycle (Fig. 6e, Eq. 1). If chloride anion was generated during the reaction, it can be easily trapped with silver nitrate to give a white precipitate of silver chloride (Fig. 6e, Eq. 2). Indeed, the white precipitate of silver chloride was detected from the standard reaction with dichloromethane as solvent. Another control reaction was done with potassium superoxide in the presence of silver nitrate and DCM as the solvent (Fig. 6e, Eq. 3). The outcome of the

reaction also shows a characteristic white precipitate of silver chloride (see Supplementary Information for more details, Section 3.5).

To further consolidate this observation, it was subjected to powder X-ray diffraction analysis to conclude the presence of chloride anion in the process (Fig. 7a, see Supplementary Information for more details, Section 3.6). All the peaks that appear in the X-ray diffraction pattern (Fig. 7a) are in correspondence with that of cubic silver chloride (ICSD No. 64734)[44]. To examine the substituent effect for silanol synthesis, a Hammett plot[45] analysis was done by calculating the rates for individual substrates with different types of *p*-substituents on the aryl group. The Hammett plot ($\log(k_X/k_H)$ versus $\sigma$) exhibited a linear correlation plot with a $\sigma$ value of −0.12 and good linearity implies that oxidation proceeds through a single radical mechanism (Fig. 7b, see Supplementary Information for more details, Section 5.1). The Hammett plot suggests a single radical mechanism for oxidation if the plot shows a linear relationship between the reaction rate and the substituent constant $\sigma$. It indicates that the rate-determining step

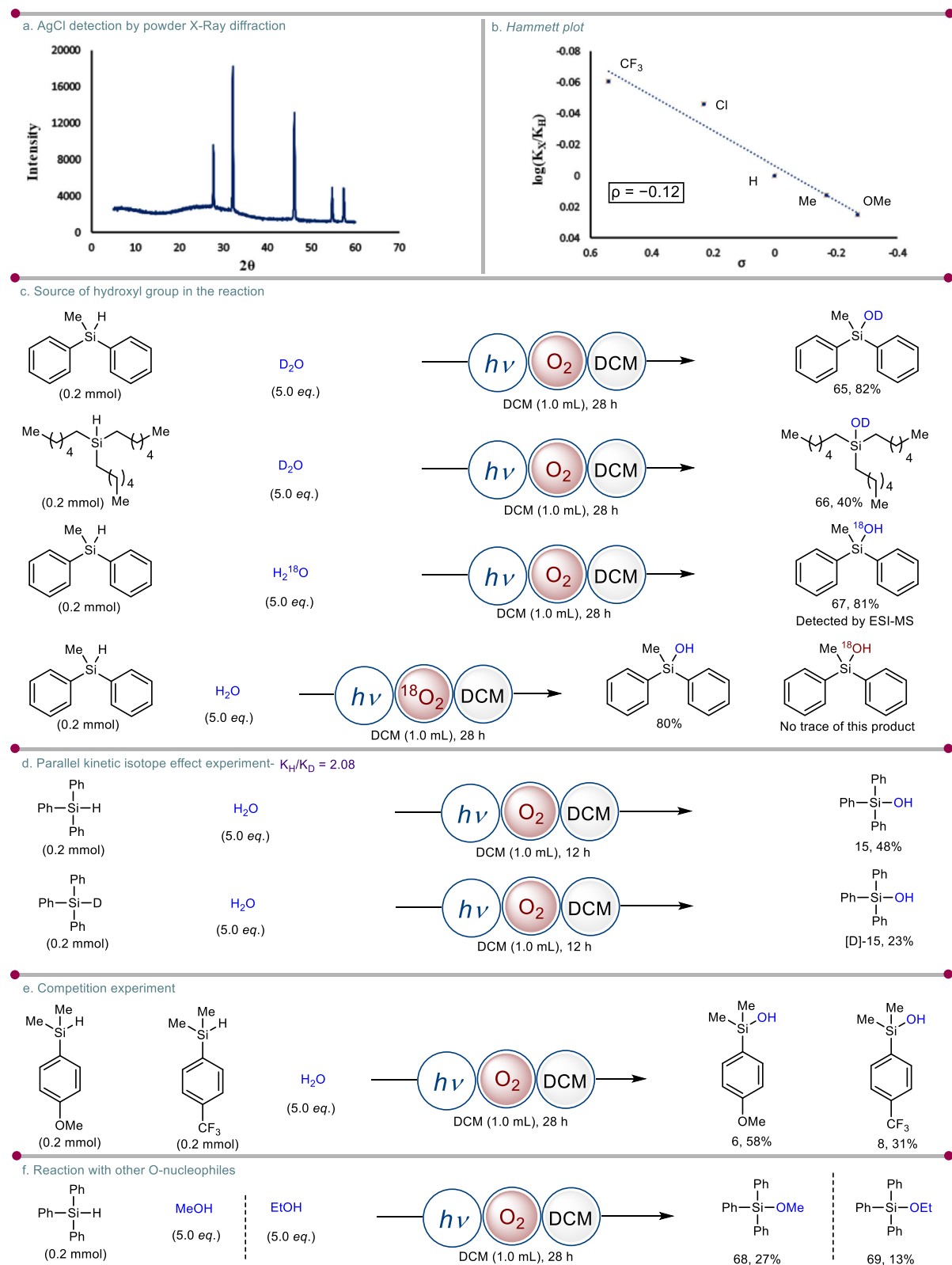

**Fig. 7 | Mechanistic aspects of photoinduced silanol synthesis. a** Powder X-ray diffraction. **b** Kinetic studies. **c** Source of the hydroxyl group in the reaction. **d** Parallel kinetic isotope effect experiment. **e** Competition experiment. **f** Reaction with other O-nucleophiles. For experimental details, see Supplementary Methods.

involves the generation and stabilization of radicals during the oxidation process. Hence it is directly influenced by the electronic properties of the substituents. This is typically observed in radical reactions, where the reaction involves the formation and stabilization of radicals. This is consistent with a single radical mechanism. A parallel kinetic

isotope effect (KIE) study was conducted using a mixture of non-deuterated and deuterated substrates and the value of $k_H/k_D$ obtained is 2.08. This result indicated that Si−H bond cleavage might be involved in the rate-determining step of the reaction (Fig. 7d, see Supplementary Information for more details, Section 5.2). To

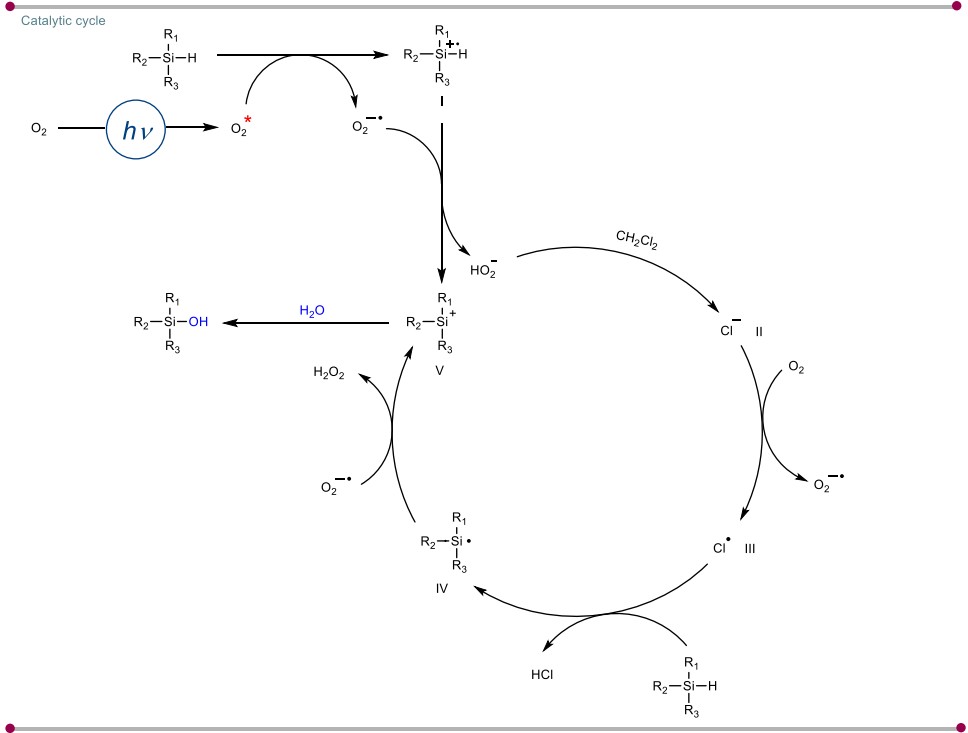

**Fig. 8 | Plausible mechanism.** Overall mechanistic cycle of the photoinduced silanol synthesis.

check the source of the hydroxyl group in silanol formation, two independent reactions were carried out in the presence of $D_2O$ and $H_2{}^{18}O$, separately. The deuterated (**65–66**) and $^{18}O$-labeled silanol (**67**) were observed which confirmed that $H_2O$ is the source of the hydroxyl group in the silanol products (Fig. 7c). The adduct **64** and $^{18}O$-labeled silanol **67** were detected by ESI-MS analysis (see Supplementary Information for more details, Section 3.7). We have performed another control experiment with $^{18}O_2$. The outcome of this reaction shows there is no formation of the $^{18}O$-labeled (**68**) product. This further confirms that $H_2O$ is the source of the hydroxyl group in the silanol products, unlikely from the oxygen present in the reaction medium (Fig. 7c, see Supplementary Information for more details, Section 3.8).

Besides this, a competition experiment was performed highlighting that electron-rich silanes undergo faster conversion to silanols compared to electron-withdrawing silanes (Fig. 7e, see Supplementary Information for more details, Section 5.3). Lastly, we tried photoinduced alcoholysis of triphenylsilane with MeOH and EtOH as ether sources. Gratifyingly, the silyl ethers were formed in both cases with decent yields (**68–69**). These results further verify the intermediacy of a silyl cation in our reaction condition (Fig. 7f). Apart from this, the conversion profile of the reaction was monitored through an on/off experiment (see Supplementary Information for more details, Section 4). The product formation ceased when the blue LED was switched off, indicating the requirement of constant photo irradiation for an effective transformation. In addition, the quantum yield of the reaction was also determined and it was found to be 6.08 (see Supplementary Information for more details, Section 3.5). This observation suggests that the reaction is going via a radical chain mechanism.

Based on the outcome of the mechanistic investigations, a plausible mechanism was depicted in (Fig. 8). The initial step involved the generation of singlet oxygen species from molecular oxygen in the presence of visible light. Consequently, the singlet oxygen ($^1O_2$) will abstract an electron from silane, leading to the formation of a silyl radical cation **I** and superoxide anion[46,47]. Then superoxide anion is converted to a very reactive hydroperoxide ion by converting silyl

radical cation **I** to silyl cation. After this, hydroperoxide ion ($HO_2{}^-$) will provide the chloride anion **II** from chlorinated solvent via nucleophilic substitution or via an electron-transfer pathway[48,49]. The chloride anion further led to chlorine radical (Cl•) **III** via the SET mechanism. The chlorine radical (Cl•) readily abstracts the hydrogen atom from the Si–H bond to generate the silyl radical **IV**, which is further oxidized by superoxide to produce a silyl cation **V**. Finally, the silyl cation **V** undergoes nucleophilic attack by $H_2O$ to furnish the desired silanol product. In the penultimate step, we hypothesized that $H_2O_2$ was formed from superoxide species. Hence, there is a probability that $H_2O_2$ can also assist in silanol formation. To check this, we charged $H_2O_2$ in the reaction mixture devoid of oxygen atmosphere, and no product formation was observed. This clearly suggests that hydrogen peroxide has no role in the product formation or oxidation process. However, another pathway is possible where the proton can be abstracted by the superoxide radical anion to give hydroperoxy radical (HOO•) and a silyl radical[50]. Finally silyl radical and (HOO•) radical can combine to give silyl peroxide, which will give the silanol product. However, the control experiment with $H_2{}^{18}O$ and $^{18}O_2$ shows no product formation, suggesting and validating the proposed reaction mechanism.

## Conclusion

In summary, we have devised a general and practical photochemical protocol for the oxidation of hydrosilanes, providing a wide array of synthetically valuable silanols. The present strategy renders streamlined synthesis of bio-relevant organosilanols via late-stage functionalization. The photochemical transformation can also be merged with flow chemistry to achieve a highly scalable silanol synthesis. It is further demonstrated that the chlorine radical (Cl•) can be generated from a common chlorinated solvent which is responsible for the silyl radical formation. Such a pathway turns out to be an elusive form of chemistry in the genre of photochemical reaction. The non-requirement of metal catalysis or stoichiometric oxidant under the photochemical condition is sufficient to effectuate this sustainable transformation. This transformation possesses general and scalable

features that are expected to be utilized for wide-scale industrial and biological applications.

## Methods

### Photoinduced synthesis of silanols

An oven-dried screw-capped reaction tube equipped with a magnetic stir-bar was charged with corresponding solid silanes (0.2 mmoL, 1 *equiv.*). Then, the tube was capped with a screw cap with a rubber septum and purged with oxygen three times. Then, the cap was wrapped with a teflon. Subsequently, water (1.0 mmoL, 5 *equiv.*) and 1 mL of dichloromethane were added to the reaction tube. Then the reaction tube was placed 3 cm away from the 34 W Kessil lamp with stirring (1000 rpm) for 28 h. For liquid silanes, the addition of silanes was done after the addition of the reaction solvent. The temperature was maintained by cooling with two fans. Upon completion of the reaction, the solvent was removed under reduced pressure, and the crude mixture was purified by column chromatography.

## Data availability

All data supporting the findings of this study including experimental procedures and compound characterization are available in the Supplementary Information. All the other data are available from the corresponding author upon request.

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

## Acknowledgements

Financial support received from SERB (CRG/2022/004197) is gratefully acknowledged. The authors are thankful for the financial support from Bristol Myers Squibb, USA. The authors thank CSIR-India (fellowship to A.S) and IIT Bombay (W.A.) for financial support.

## Author contributions

A.S., W.A. and D.M. conceived the concept. A.S. and W.A. performed the reactions. A.S., W.A., D.B.W. and D.M. designed the control experiments, mechanistic pathway, and analysis of the product. A.S., W.A., D.B.W. and D.M. wrote the paper.

## Funding

## Competing interests

The authors declare no competing interests.
