## [Peer Review File · Nature Communications]

Highly Scalable Photoinduced Synthesis of Silanols via Untraversed Pathway for Chlorine Radical ($\text{Cl}\bullet$) GenerationReviewers' Comments:

Reviewer #1:

Remarks to the Author:

In this work, Maiti and co-workers report photoinduced oxidation of hydrosilanes to silanols by employing water as the hydroxy source under aerobic conditions. This protocol provides a broad range of silanols with good functional group compatibility, and it is applicable to the late-stage functionalization of complex hydrosilanes. The scalable silanol synthesis using the photochemical flow method and the synthetic application of the current protocol are also demonstrated by the authors. However, given the fact that the direct transformation of hydrosilanes into silanols by oxidation of Si–H bonds has been intensively investigated (related references are insufficiently cited in the text), including the metal-free visible-light-driven methods (see *Chem. Commun.* 2023, 59, 6588; *Sci. China Chem.* 2018, 61, 1594; not cited in the text), this work does not make significant advance compared with those reported examples. Although the plausible mechanistic scenario involving a chlorine radical intermediate in situ generated from chlorinated solvent (CH₂Cl₂) may be somewhat interesting, the proposed catalytic cycle (Fig. 8 in the main text) seems unreasonable and confusing (described in detail below). Hence, both the novelty and the quality of this work are considered not high enough for publication in *Nat. Commun.*

Other comments and questions:

1) My major concerns lie with the reaction mechanism.

First, the authors ascribed the initial silyl radical generation to the homolytic cleavage of Si–H bonds in hydrosilanes only with the help of visible-light irradiation (blue LEDs in this work, wavelength > 400 nm, energy < 71 kcal/mol), which is illogical considering the bond dissociation energies (BDEs) of related Si–H bonds in this work (around 85–90 kcal/mol; see: Luo Y.-R., *Handbook of Bond Dissociation Energies in Organic Compounds*, 2002, CRC Press, <https://doi.org/10.1201/9781420039863>). In this aspect, the hydrogen atom transfer (HAT) of hydrosilanes with a certain radical species might be the most likely mechanism (for example, see *ACIE* 2021, 60, 1839; *ACIE* 2020, 59, 15507). In addition, the control experiment with KO₂ in the absence of irradiation should be carried out since the proposed mechanism relied on irradiation for superoxide formation.

Second, the authors are suggested to calculate the quantum yield of the reaction to verify whether there is a radical chain or not. The simple on-off control experiment with respect to light irradiation is not a piece of decisive evidence in this aspect (*Chem. Sci.* 2015, 6, 5426.).

Third, the control experiments with deuterated water are meaningless since silanol products have acidic hydroxy groups and they can undergo H/D exchange with the added deuterated water.

Fourth, an additional control experiment with ¹⁸O₂ is necessary to fully exclude the possible source of hydroxy groups from O₂. In addition, the reaction proceeded well in the absence of any externally added water (57% yield) according to the Supplementary Information. In this sense, the use of ¹⁸O₂ should be mandatory to clarify the source of the hydroxy group. Thus, a control experiment with ¹⁸O₂ in the absence of externally added water may be necessary.

Last but not least, the claimed important role of chlorine radical species requires solid evidence. Nucleophilic substitution reactions of superoxide with CH₂Cl₂ may generate chloride anions, but at the same time, peroxy radicals are also formed, which itself or other derived radicals may abstract hydrogen atoms from hydrosilanes instead of chlorine radicals. In this aspect, the authors are suggested to conduct control experiments with solvents that generate bromide and iodide anions. If the reaction can still proceed, the authors should revise the mechanism accordingly. In addition, further control experiments with tetrabutylammonium salts other than chloride ones should be taken

to exclude any contribution from the cation.

Overall, further experimental (and computational?) studies to show more pieces of evidence for the proposed reaction mechanism are highly suggested (for example, see: Nat. Commun. 2021, 12, 4010).

2) How about other types of nucleophiles, such as alkyl alcohols and phenols? These results, if successfully achieved, could not only further verify the intermediacy of a silyl cation but also show the potential of the current photochemical strategy to afford silyl ethers under mild conditions.

3) The authors should note that the graphic presentation of this work is largely unreadable, which makes the readers very hard to follow. Particularly, the detailed reaction conditions in Figs. 5–7 are missing and the corresponding footnotes are confusing. To clearly present these results, the authors are suggested to directly provide more detailed information in these figures. In addition, a brief optimization table of reaction conditions is better to be provided in the main text.

Reviewer #2:

Remarks to the Author:

In this manuscript, Maiti et al. reported a highly scalable photoinduced synthesis of silanols, including diversely functionalized organosilanols, silanediols, and polymeric siloxanol, engaging a wide spectrum of hydrosilanes under ambient reaction conditions. In addition, the streamlined synthesis of bio-active silanols via late-stage functionalization was also carried out. Interestingly, this work revealed an interesting example of photoinduced non-classical chlorine radical ($\text{Cl}\bullet$) generation from a readily available chlorinated solvent under aerobic conditions. In my opinion, this work is suitable for publication on Nature Communications after minor revision as following:

(1) The yields showed in this work could be clarified clearly as isolated yields? NMR yields? Especially for unstable silanol 37, how to get the pure product?

(2) In the Introduction part, about the history of silanol synthesis, more reasonable descriptions should be added for the synthetic methods reported in the past decades.

(3) DCM play important role in this reaction and could be recognized as a chlorine radical precursor, how about other alkyl chlorides, such as DCE, CHCl_3 or its analogues.

Reviewer #3:

Remarks to the Author:

Maiti and coworkers reported photoinduced synthesis of a wide array of synthetically valuable silanols from silanes promoted by chlorinated solvent. The reaction proceeds under mild metal-free conditions, with O_2 and H_2O as green oxidant and reagent. The photochemical transformation can also be merged with flow chemistry to achieve a highly scalable silanol synthesis. The mechanism study indicates the necessity of light, O_2 and chlorinated solvent. This paper is overall well presented, however, I have some concerns about the importance of this work. As the author said in background part, chlorine radical could be generated by inorganic chloride salts, so what's the advantage by using chlorinated solvent as chlorine radical source? The large excess of chlorinated solvent and photo-catalyst free condition may make the chlorine radical generation uncontrollable. Besides, the author claims that "The implementation of chlorine radical ($\text{Cl}\bullet$) to generate heteroatom-centered radicals remains a major challenge in organic chemistry", but in ref. 15, there are several silane substrates undergoing chlorine radical mediated HAT reaction to generate Si-centered radical. I don't think there exist such challenges. Therefore, I suggest the author make a major revision of this paper. Below are some detailed comments.

The author proposed that the silyl radical was initially generated through homolytic cleavage of

silanes. Does that mean H₂ release could be observed when silanes were irradiated by light in the absence of O₂? Does the (TMS)₃SiH have any absorption within blue light region? Has the author ever considered the possibility of SET process between silane and excited O₂, generating Si-radical cation and superoxide radical anion? I think the mechanism study part needs further exploration.

Some reference citations make me confused. For example, "Similarly, tetrabutyl-ammonium chloride (TBACl) can serve as a chlorine radical source under photo-irradiation in the presence of CeCl₃ for a cross-coupling reaction (Fig. 1c).¹³" But in ref 13, only Co-catalyst used, no CeCl₃ was added at all! Another case is, "We envisaged that chlorinated solvent under photochemical conditions will provide the rudimentary chlorine radical (Cl•).^{13,15}" I don't know why the author cites ref 13 and 15 here, is there any relationship between these two references with this sentence? I suggest the author carefully check whether the cited references are related and support the sentence.

Could author explain in detail why does the Hammett plot suggest that the oxidation proceeds through a single radical mechanism ?

Response letter

Reviewer #1 (Comments for the Author): In this work, Maiti and co-workers report photoinduced oxidation of hydrosilanes to silanols by employing water as the hydroxy source under aerobic conditions. This protocol provides a broad range of silanols with good functional group compatibility, and it is applicable to the late-stage functionalization of complex hydrosilanes. The scalable silanol synthesis using the photochemical flow method and the synthetic application of the current protocol are also demonstrated by the authors. However, given the fact that the direct transformation of hydrosilanes into silanols by oxidation of Si–H bonds has been intensively investigated (related references are insufficiently cited in the text), including the metal-free visible-light-driven methods (see Chem. Commun. 2023, 59, 6588; Sci. China Chem. 2018, 61, 1594; not cited in the text), this work does not make significant advance compared with those reported examples. Although the plausible mechanistic scenario involving a chlorine radical intermediate in situ generated from chlorinated solvent (CH_2Cl_2) may be somewhat interesting, the proposed catalytic cycle (Fig. 8 in the main text) seems unreasonable and confusing (described in detail below). Hence, both the novelty and the quality of this work are considered not high enough for publication in Nat. Commun.

Our response: We are thankful to the reviewer for his/her constructive suggestions regarding the manuscript. We have made revisions/addressed the queries and necessary modifications have been included in the main manuscript.

Comment: 1) My major concerns lie with the reaction mechanism. First, the authors ascribed the initial silyl radical generation to the homolytic cleavage of Si–H bonds in hydrosilanes only with the help of visible-light irradiation (blue LEDs in this work, wavelength > 400 nm, energy < 71 kcal/mol), which is illogical considering the bond dissociation energies (BDEs) of related Si–H bonds in this work (around 85–90 kcal/mol; see: Luo Y.-R., Handbook of Bond Dissociation Energies in Organic Compounds, 2002, CRC Press, <https://doi.org/10.1201/9781420039863>). In this aspect, the hydrogen atom transfer (HAT) of hydrosilanes with a certain radical species might be the most likely mechanism (for example, see ACIE 2021, 60, 1839; ACIE 2020, 59, 15507). In addition, the control experiment with KO_2 in the absence of irradiation should be carried out since the proposed mechanism relied on irradiation for superoxide formation.

Our response: 1. We are thankful to the reviewer for his/her helpful suggestions regarding the mechanism. We have performed a series of experiments to prove our mechanistic hypothesis. Finally, the experimental result shows slight modification only in the initial step of the mechanism. The modified version of the catalytic cycle indicates that the initial step involved the generation of singlet oxygen species from molecular oxygen in presence of visible light. Consequently, the singlet oxygen ($^1\text{O}_2$) will abstract an electron from silane, leading to the formation of a silyl radical cation **I** and superoxide anion. Then superoxide anion is converted to very reactive hydroperoxide ion by converting silyl radical cation **I** to silyl cation. After this, hydroperoxide ion (HO_2^-) will provide the chloride anion **II** from chlorinated solvent *via* nucleophilic substitution or *via* an electron-transfer pathway. After this the mechanism will be same as previously shown in the main manuscript.

2. The control experiment with KO_2 in the absence of irradiation shows no product formation. It suggests that light is necessary for product formation. The experiment is included in the revised manuscript and supporting information (see supporting information section 3, subsection 3.3).

Comment: Second, the authors are suggested to calculate the quantum yield of the reaction to verify whether there is a radical chain or not. The simple on-off control experiment with respect to light irradiation is not a piece of decisive evidence in this aspect (Chem. Sci. 2015, 6, 5426.).

Our response: The quantum yield of the reaction was 6.08. The calculation part is included in the revised supplementary information (see supporting information section 3, subsection 3.5) and it is also mentioned in the main manuscript.

Comment: Third, the control experiments with deuterated water are meaningless since silanol products have acidic hydroxy groups and they can undergo H/D exchange with the added deuterated water.

Our response: In the previous literature reports this type of control experiment are generally done to prove the source of water. We want to highlight another experiment which is already given in the manuscript with H_2^{18}O (see figure 7, part c and supporting information section 3, subsection 3.8). The formation of H_2^{18}O silanol suggests that hydroxyl group coming from water. In addition, we have performed another control experiment with $^{18}\text{O}_2$ as suggested in the 4th comment. The outcome of the result shows non labelled silanol product. There is no formation of the ^{18}O -labelled (**68**) product. This further consolidate that H_2O is the source of the hydroxyl group in the silanol products, unlikely from the oxygen present in the reaction medium.

Comment: Fourth, an additional control experiment with $^{18}\text{O}_2$ is necessary to fully exclude the possible source of hydroxy groups from O_2 . In addition, the reaction proceeded well in the absence of any externally added water (57% yield) according to the Supplementary Information. In this sense, the use of $^{18}\text{O}_2$ should be mandatory to clarify the source of the hydroxy group. Thus, a control experiment with $^{18}\text{O}_2$ in the absence of externally added water may be necessary.

Our response: Thank you for this valuable suggestion. We have performed this control experiment with $^{18}\text{O}_2$ to fully exclude the possible source of hydroxy groups from O_2 . The outcome of this reaction shows non labelled silanol product. There is no formation of the ^{18}O -labelled (**68**) product. This further consolidate that H_2O is the source of the hydroxyl group in the silanol

products, unlikely from the oxygen present in the reaction medium. This part is included in the revised supplementary information (see supporting information section 3, subsection 3.8) and main manuscript.

Comment: Last but not least, the claimed important role of chlorine radical species requires solid evidence. Nucleophilic substitution reactions of superoxide with CH_2Cl_2 may generate chloride anions, but at the same time, peroxy radicals are also formed, which itself or other derived radicals may abstract hydrogen atoms from hydrosilanes instead of chlorine radicals. In this aspect, the authors are suggested to conduct control experiments with solvents that generate bromide and iodide anions. If the reaction can still proceed, the authors should revise the mechanism accordingly. In addition, further control experiments with tetrabutylammonium salts other than chloride ones should be taken to exclude any contribution from the cation. Overall, further experimental (and computational?) studies to show more pieces of evidence for the proposed reaction mechanism are highly suggested (for example, see: Nat. Commun. 2021, 12, 4010).

Our response: **1.** In the manuscript it is already mentioned the peroxy species has no role in reaction. **2.** The control experiment with solvents that generate bromide and iodide anions (bromobenzene and chlorobenzene as solvent) are previously mentioned in the manuscript as well as in supporting information (see supporting information section 2.2, Table S4). We have also tested iodobenzene as a solvent and result shows trace amount of silanol product. It suggests that chlorine radical plays an integral role in the reaction. **3.** The control experiment with tetrabutylammonium salts other than chloride was performed with tetrabutylammonium bromide, tetrabutylammonium iodide and tetrabutylammonium fluoride shows yield of 7, 8 and 3% respectively (see supporting information section 3, subsection 3.6). **4.** A series of control experiments was previously performed and also new experiments was also included to strengthen the mechanism.

Comment: 2) How about other types of nucleophiles, such as alkyl alcohols and phenols? These results, if successfully achieved, could not only further verify the intermediacy of a silyl cation but also show the potential of the current photochemical strategy to afford silyl ethers under mild conditions.

Our response: We are thankful to the reviewer for this constructive suggestion. We have checked our protocol with a series of nucleophiles such as methanol, ethanol, *t*-butanol, isopropanol, butanol, phenol, 4-methoxy phenol and 4-chloro phenol. But among these only methanol and ethanol give ether product in 27% and 13% respectively (see figure 7, part f and supporting information section 6). These silyl ethers formation further consolidate that our mechanism goes *via* silyl cation formation.

Comment: 3) The authors should note that the graphic presentation of this work is largely unreadable, which makes the readers very hard to follow. Particularly, the detailed reaction conditions in Figs. 5–7 are missing and the corresponding footnotes are confusing. To clearly present these results, the authors are suggested to directly provide more detailed information in these figures. In addition, a brief optimization table of reaction conditions is better to be provided in the main text.

Our response: **1.** The reaction conditions in Figs. 5–7 has been incorporated in the revised manuscript. **2.** A brief optimization table has been incorporated in the revised manuscript as Table 1.

Reviewer #2 (Comments for the Author): In this manuscript, Maiti et al. reported a highly scalable photoinduced synthesis of silanols, including diversely functionalized organosilanols, silanediols, and polymeric siloxanol, engaging a wide spectrum of hydrosilanes under ambient reaction conditions. In addition, the streamlined synthesis of bio-active silanols via late-stage functionalization was also carried out. Interestingly, this work revealed an interesting example of photoinduced non-classical chlorine radical ($\text{Cl}\cdot$) generation from a readily available chlorinated solvent under aerobic conditions. In my opinion, this work is suitable for publication on Nature Communications after minor revision as following:

Our response: We are thankful to the reviewer for his/her appreciation regarding the manuscript. We have made necessary modifications to improve the overall quality of the work.

Comment: (1) The yields showed in this work could be clarified clearly as isolated yields? NMR yields? Especially for unstable silanol 37, how to get the pure product?

Our response: **1.** The yields showed in this work referred as isolated yields and it is included in the revised manuscript. **2.** Yes reviewer rightly pointed about of the sensitive nature of the silanol 37. It is isolated *via* flash column chromatography to avoid decomposition.

Comment: (2) In the Introduction part, about the history of silanol synthesis, more reasonable descriptions should be added for the synthetic methods reported in the past decades.

Our response: The introduction has been modified and have been included in the main manuscript.

Comment: (3) DCM play important role in this reaction and could be recognized as a chlorine radical precursor, how about other alkyl chlorides, such as DCE, CHCl_3 or its analogues.

Our response: The other analogues of DCM such as DCE, CHCl₃ already included previously in supporting information (see supporting information section 2.2, Table S4).

Reviewer #3 (Comments for the Author): Maiti and coworkers reported photoinduced synthesis of a wide array of synthetically valuable silanols from silanes promoted by chlorinated solvent. The reaction proceeds under mild metal-free conditions, with O₂ and H₂O as green oxidant and reagent. The photochemical transformation can also be merged with flow chemistry to achieve a highly scalable silanol synthesis. The mechanism study indicates the necessity of light, O₂ and chlorinated solvent. This paper is overall well presented; however, I have some concerns about the importance of this work. As the author said in background part, chlorine radical could be generated by inorganic chloride salts, so what's the advantage by using chlorinated solvent as chlorine radical source? The large excess of chlorinated solvent and photo-catalyst free condition may make the chlorine radical generation uncontrollable. Besides, the author claims that "The implementation of chlorine radical (Cl•) to generate heteroatom-centered radicals remains a major challenge in organic chemistry", but in ref. 15, there are several silane substrates undergoing chlorine radical mediated HAT reaction to generate Si-centered radical. I don't think there exist such challenges. Therefore, I suggest the author make a major revision of this paper. Below are some detailed comments.

Our response: The reviewer's remarks were extremely encouraging and the constructive suggestions helped us immensely in improving the quality of this manuscript.

Comment: The author proposed that the silyl radical was initially generated through homolytic cleavage of silanes. Does that mean H₂ release could be observed when silanes were irradiated by light in the absence of O₂? Does the (TMS)₃SiH have any absorption within blue light region? Has the author ever considered the possibility of SET process between silane and excited O₂, generating Si-radical cation and superoxide radical anion? I think the mechanism study part needs further exploration.

Our response: The reviewer's suggestions are very helpful and constructive towards the reaction mechanism. We have performed a series of experiments to prove our mechanistic hypothesis. **A.** We have done a control experiment with triphenylsilane in presence of visible light but in the absence of O₂. This experiment shows no hydrogen evolution during the reaction. **B.** According to the suggestion the SET process between silane and excited O₂ was evaluated. In order to confirm the presence of singlet oxygen we conducted trapping experiment with anthracene as the trapping agent. This resulted in the formation of the endoperoxide product (Fig. 6c, see supporting information for more details, Section 3.3). Importantly, this endoperoxide product was not observed in reactions conducted in the dark conditions. This provides compelling evidence supporting the formation of singlet oxygen within our reaction system.

Finally, the experimental result shows slight modification only in the initial step of the mechanism. The modified version of the catalytic cycle indicates that the initial step involved the generation of singlet oxygen species from molecular oxygen in presence of visible light. Consequently, the singlet oxygen (¹O₂) will abstract an electron from silane, leading to the formation of a silyl radical cation **I** and superoxide anion. Then superoxide anion is converted to very reactive hydroperoxide ion by converting silyl radical cation **I** to silyl cation. After this, hydroperoxide ion (HO₂⁻) will provide the chloride anion **II** from chlorinated solvent *via* nucleophilic substitution or *via* an electron-transfer pathway. After this the mechanism will be same as previously shown in the main manuscript.

Comment: Some reference citations make me confused. For example, "Similarly, tetrabutyl-ammonium chloride (TBACl) can serve as a chlorine radical source under photo-irradiation in the presence of CeCl₃ for a cross-coupling reaction (Fig. 1c).¹³" But in ref 13, only Co-catalyst used, no CeCl₃ was added at all! Another case is, "We envisaged that chlorinated solvent under photochemical conditions will provide the rudimentary chlorine radical (Cl•).^{13,15}" I don't know why the author cites ref 13 and 15 here, is there any relationship between these two references with this sentence? I suggest the author carefully check whether the cited references are related and support the sentence.

Our response: We are thankful to the reviewer for this valuable suggestion. There is a typo error for the reference 13, so it is replaced with appropriate reference. Similarly, after further analysis references 15 also has been updated in the revised manuscript (the reference 13 and 15 are now 15 and 17 in the revised manuscript).

Comment: Could author explain in detail why does the Hammet plot suggest that the oxidation proceeds through a single radical mechanism?

Our response: The detailed explanation of the Hammet plot has been incorporated in the revised manuscript.

Sincerely,
Deb
Debabrata Maiti

Reviewers' Comments:

Reviewer #1:

Remarks to the Author:

In the revised version of the manuscript, the authors have made a great effort to address all of concerns of referees. Now, this manuscript is suitable for publication in Nat. Commun.

Reviewer #2:

Remarks to the Author:

The revised manuscript has been improved and can be accepted for publication on NC.

Reviewer #3:

Remarks to the Author:

As most of the reviewer's comments have been addressed and corresponding revisions have been made in the manuscript, I support publication of this paper in Nature Communication.

Submission of Revised Manuscript NCOMMS-23-33981A-Z

Reviewer #1 (Comments for the Author): In the revised version of the manuscript, the authors have made a great effort to address all of concerns of referees. Now, this manuscript is suitable for publication in Nat. Commun.

Our response: We are thankful to the reviewer that he/she finds suitable for publication in *Nature Communications* after revision.

Reviewer #2 (Comments for the Author): The revised manuscript has been improved and can be accepted for publication on NC.

Our response: We are thankful to the reviewer for accepting the manuscript to publish in *Nature Communications*.

Reviewer #3 (Comments for the Author): As most of the reviewer's comments have been addressed and corresponding revisions have been made in the manuscript, I support publication of this paper in *Nature Communication*.

Our response: We are thankful to this reviewer for supporting publication of this paper in *Nature Communication*.

I sincerely hope that you will find this manuscript suitable for wide readership of *Nature Communication*.

Sincerely,
Deb
Debabrata Maiti